# The Effect of High and Low Life Purpose on Ikigai (a Meaning for Life) among Community-Dwelling Older People—A Cross-Sectional Study

**DOI:** 10.3390/geriatrics6030073

**Published:** 2021-07-24

**Authors:** Souma Tsuzishita, Tadaaki Wakui

**Affiliations:** 1Department of Rehabilitation, Faculty of Health Science, Nara Gakuen University, Nara 631-8524, Japan; 2Faculty of Health and Well-Being, Kansai University, Osaka 590-8515, Japan; t.wakui@kansai-u.ac.jp

**Keywords:** care prevention, life purpose, ikigai

## Abstract

The purpose of this study is to reveal how high or low life purpose is related to QOL and ikigai (a meaning for life). Ikigai is “a sense of purpose and motivation in the daily lives of older people, a sense that they are capable and meaningful to their families and others, and that they should be”. Eighty-one community-dwelling older people (23 male and 58 female, mean age 77 ± 5.2 years) participated of their own will. The following items were measured: dementia test, exercise habits, life purpose, ikigai, and QOL. In the multivariate analysis of high and low life purpose, only ikigai was found to be related. In the multivariate analysis of ikigai, life purpose was also the most relevant, indicating that life purpose and ikigai are strongly interrelated. To improve QOL, it is also necessary to take into account life purpose in addition to the current nursing care prevention.

## 1. Introduction

In Japan, the gap between average life expectancy and healthy life expectancy is becoming an issue as the population ages. To extend healthy life expectancy, long-term care prevention programs are being widely implemented throughout the country. The Ministry of Health, Labor and Welfare (MHLW) [1] in Japan has published a manual on long-term care prevention, which states that prevention should not only aim at improving individual factors such as the motor function and nutritional status of older people, but also the quality of life (QOL) by supporting individuals in achieving their “ikigai” (a meaning for life) and self-actualization. In other words, it is important to understand that the goal of nursing care prevention is to improve the quality of life in nursing care prevention. Spirituality, such as ikigai and self-actualization, is also essential in practicing nursing care prevention.

According to MHLW [2], the World Health Organization (WHO) considers spirituality crucial for human dignity and quality of life. WHO also states in its definition of health that spirituality, in all the physical, psychological, and social domains, is a vital part of the whole person [2]. Although the proposed revision of this definition of health is still pending [2], spirituality is undoubtedly important for human health. Fujii [3] states that the common elements of spirituality among Japanese people include the meaning and purpose of life and living. Therefore, it should be considered that owning a sense of meaning and life purpose positively affects the health of older people.

Alimujiang et al. [4] defined life purpose as “a self-organized life orientation that stimulates goals” and reported that having a life purpose may promote healthy behaviors and give meaning to one’s life. Boyle et al. [5] investigated the effects of high and low life purpose on Activities of Daily Living (ADL) impairment, Instrumental Activities of Daily Living (IADL) impairment, and mobility impairment in community-dwelling older people. They report that having a higher purpose in life is associated with a reduced risk of ADL impairment, IADL impairment, and mobility impairment. Higher life purpose was also associated with a lower risk of mortality [6,7], cognitive dysfunction [8,9], and physical pain [10]. Thus, there are many reports that the level of life purpose is closely related to health. However, there are only a limited number of reports on the relationship between high and low life purpose and mental health [11]. Hill et al. [11] investigated the effects of high and low life purpose on changes in daily physical symptoms, positive and negative emotions related to stress, and found that people with high life purpose had higher positive emotions and lower negative emotions. In addition, those with high life purpose encountered the same number of daily stressors as those with low life purpose, but less increase in negative emotions and physical symptoms (headache, fatigue, cough, etc.) is reported on stressful days than on non-stressful ones [11]. In this study, only positive or negative emotions were investigated in terms of mental health. Yet, QOL and ikigai are also important when considering the mental health of older people. As mentioned, MHLW [1] has published a manual on nursing care prevention, which states that nursing care prevention aims at improving QOL by supporting older people to achieve ikigai and self-actualization.

To conclude, life purpose affects physical aspects such as ADL impairment, IADL impairment, mobility impairment, mortality risk, cognitive dysfunction, and physical pain. Moreover, overseas studies have shown that life purpose reduces stress and is associated with negative emotions. Nonetheless, to the best of our knowledge, the number of reports on the effects of high and low life purpose on the mental health of older people in Japan is very much limited [11,12,13]. Moreover, when assessing the mental health of older people in Japan, the evaluation of QOL and ikigai from the perspective of care prevention is also, undoubtedly, critical. In one of our previous studies, the QOL and ikigai of the high and low life purpose groups were compared, and the values for physical health, mental health, and ikigai were higher in the high life purpose group than in the low group [12]. In another study, the high life purpose group showed significantly higher values for mental health and ikigai than the low life purpose group [13]. These results indicate that high or low life purpose is associated with QOL and ikigai, but these studies had small samples and multivariate analyses were not conducted to verify the strength of the association. Furthermore, QOL in these studies used outcomes such as PCS (physical component summary), which is a physical summary score (physical functioning, physical role, bodily pain, etc.), and MCS (mental component summary), which is a mental summary score (social functioning, emotional role, mental health, etc.) calculated by weighting and adding the scores of each item of the Short Form-8 (SF-8). Hence, the relationship between the values of each of the SF-8 subscales (physical functioning, physical role, bodily pain, general health, vitality, social functioning, emotional role, and mental health) and the level of life purpose cannot indeed be revealed.

The purpose of this study is, thus, to examine the strength of the association by increasing the sample size and conducting multivariate analysis to reveal how high or low life purpose is related to QOL and ikigai.

## 2. Materials and Methods

### 2.1. Subjects

Eighty-one older people (23 male and 58 female, mean age 77 ± 5.2 years) were recruited from community centers in Osaka City (Osaka Prefecture), and Nara City (Nara Prefecture). The survey was conducted from 23 August 2019, to 5 December 2019.

### 2.2. Data Collection

The items measured were the dementia test (Test your memory, Japanese version), presence of exercise habit, life purpose, ikigai, and QOL.

To ensure all our subjects had a thorough understanding of the questionnaire, cognitive function was tested using the Japanese version of Test your memory (TYM-J) [14]. The full score of TYM-J is 50 points, and the sensitivity of TYM-J is 93% for the diagnosis of Alzheimer’s disease. The mean TYM-J score of the subjects in this study was 47.4 ± 1.4 points, and therefore, no subject has dementia.

The definition of exercise habit was based on the frequency and average duration of exercise as defined in the National Health and Nutrition Examination Survey [15]. In the questionnaire, those who exercised two or more days a week and exercised for 30 min or more on average were classified as having an exercise habit, and those who did not were classified as not having an exercise habit.

For life purpose, we used the Rinro-Shiki Life and Death Scale by Hirai et al. [16], which allows us to score Japanese people’s views on life and death in seven questions (1. not true to 7. true).

The ikigai-9 [17] scale was used for ikigai. This scale is simple and easy to use with only nine items, and with five answers for each item, the total score is 45 points. As an individual outcome measure, the small number of items is important in terms of lightening the burden on the subject. The distribution of scores, reliability, and validity in healthy middle-aged and older people have been reported.

The MOS 36-Item Short-Form Health Survey (SF-36) is a typical method of measuring QOL, but it is difficult for older people for its large number of questions. In this study, we used the Short Form-8 Japanese version (SF-8 Japanese version) [18], which was developed as a shortened version based on the eight subscales of the SF-36 (physical functioning, physical role, bodily pain, general health, vitality, social functioning, emotional role, and mental health). A license agreement was obtained for the use of the Japanese version SF-8 in this study.

### 2.3. Statistical Analysis

The analysis method was as follows. First, the median was calculated because the values of the life purpose of the subjects were not normally distributed, and the subjects were classified into two groups—“5 Somewhat true to 7 True” was the high life purpose group, and “1 Not true at all to 4 Neither true nor false” was the low life purpose group. The numerical values and scores of all evaluation items were compared between the two groups using the *t*-test, χ^2^ test after cross-tabulation, and the Mann–Whitney *U* test. Second, for the assessment items that showed significant differences between the two groups in the *t*-test or χ^2^ test, correlations between the assessment items were examined by Spearman’s rank correlation coefficient to avoid multicollinearity before binomial logistic regression analysis. Third, based on the results of the *t*-test, χ^2^ test, and correlation analysis, independent variables were extracted from each assessment item, and a binomial logistic regression analysis was conducted using the forced entry method, with high and low life purpose as dependent variables. In addition, a binomial logistic regression analysis was conducted again with age, gender, and exercise habits as independent variables. Fourth, the variable with the strongest impact in the aforementioned binomial logistic regression analysis was set as the dependent variable, and independent variables were extracted from each evaluation item based on the results of the *t*-test, χ^2^ test, and correlation analysis, and multiple regression analysis by the forced entry method was conducted.

Statistical analysis software SPSS version 27 for Windows (IBM) was used for statistical processing, and the statistical significance level was set at 5%.

### 2.4. Ethical Considerations

We verbally explained the purpose and content of this study to the participants, that participation was entirely based on the participant’s free will and no disadvantage would be caused if they refuse to fill out questionnaires; they could also stop the study even after presenting their consent. No individual respondent chose not to be identified because the data would be processed statistically. The survey was conducted using self-administered questionnaires, which was a self-administered survey that lasts for about 10 to 15 min.

## 3. Results

A total of 39 out of the 81 subjects were in the high life purpose group and 42 were in the low life purpose group. Table 1 shows the results of the univariate analysis. Age, gender, exercise habits, physical functioning, physical role, bodily pain, general health, vitality, social functioning, and emotional role were similar between the two groups. On the other hand, ikigai and mental health were significantly higher in the group with high life purpose than in the group with low life purpose (*p* < 0.001, *p* < 0.05, respectively). In addition, the high life purpose group had higher mean values for all QOL items than the low life purpose group.

Next, in the univariate analysis, the results of the correlation matrix of the items that showed significant differences were judged to have low multicollinearity, because although a significant correlation (*r* = 0.433, *p* < 0.001) was found between life satisfaction and mental health in the analysis of internal correlation, the *r* did not exceed 0.8.

Next, we conducted a binomial logistic regression analysis using the forced entry method, with the two items of ikigai and mental health showing a significant difference between the high and low life purpose groups in the univariate analysis as independent variables, and “life purpose”, which was classified into the high and low life purpose groups as the dependent variable. As a result, a significant association was found only for ikigai (odds ratio: 1.156, 95% confidence interval: 1.058–1.264; *p* < 0.01; Table 2). The Hosmer-Lemeshow test was *p* > 0.05, with a target rate of 70.4%, indicating a good fit of the logistic regression model.

Furthermore, age, gender, and exercise habits were added as independent variables, and another round of binomial logistic regression analysis was performed. Nevertheless, only ikigai (odds ratio: 1.179, 95% confidence interval: 1.078–1.289) showed a significant association (*p* < 0.001; Table 3).

Finally, based on the results of the *t*-test, χ^2^ test, and correlation analysis, a multiple regression analysis using the forced entry method was conducted with age, general health, vitality, social functioning, emotional role, mental health, and life purpose as independent variables, with the most relevant variable in the binomial logistic regression analysis described above as the dependent variable. A multiple regression analysis using the forced entry method was conducted. The results showed that there was a significant relationship between life purpose (β: 0.433, *p* < 0.001) and age (β: 0.126, *p* < 0.05; R^2^ = 0.430; Table 4).

## 4. Discussion

The results in the univariate analysis for mental health and life purpose showed that there was a significant difference between high and low life purpose. In addition, the high life purpose group had higher mean values for all QOL items than the low life purpose group. In the multivariate analysis, there was a significant difference only in ikigai. When age, gender, and exercise habits were added as independent variables and for another multivariate analysis, a significant difference was found only in ikigai.

In the univariate analysis, high and low life purpose was found to be associated with mental health. In previous studies, high life purpose is associated with high daily positive emotions and low daily negative emotions [11]. Furthermore, Kobayashi et al. [19] listed resilience as something that influences health and longevity and reported that one of the components of this resilience is “life purpose,” which is associated with increased positive acceptance. We witnessed in the present study that high and low life purpose was related to positive emotions of mental health, similar to previous studies. Next, regarding the fact that high and low life purpose was associated with high values for all QOL items, Zilioli et al. [20] investigated the effect of life purpose on chronic stress 10 years later in 985 people (mean age 46.14 ± 11.7 years). The results showed that high life purpose was associated with suppression of inflammatory cytokines and other substances associated with chronic stress. In addition, people with high life purpose are more likely to have the mentality that they can control their health rather than others, which has a positive effect on their subjective health status [20]. In the present study, it can be inferred that people with high life purpose are more likely to have the mentality that they can control their health, and this may have had a positive impact on all QOL items. The results of this study were similar to previous studies, suggesting that high life purpose is an important factor in improving QOL in the future.

Next, in a multivariate analysis with high and low life purpose as the dependent variables, ikigai was most related to high and low life purpose. In addition, this study conducted a multiple regression analysis (forced entry method) with life purpose as the dependent variable, and the results showed that life purpose was most related to ikigai. This suggests that high and low life purpose are strongly interrelated with ikigai; life purpose is crucial for improving ikigai, which is emphasized in the care prevention manual.

In a study of the relationship between life purpose and ikigai, Kondo et al. [21] surveyed 162 urban homebound older people aged 60 or above, asking them to choose from 15 items related to ikigai, including “motivation and sense of purpose”, “sense of role, contribution, and usefulness”, “sense of accomplishment”, “sense of mission, responsibility, and duty”, and “sense of challenge”. As a result, “motivation and sense of purpose” was most selected. Moreover, Kondo et al. [21] defined ikigai as “the sense of motivation of older people to live with a sense of purpose and motivation in daily life, and a sense of motivation to live with the awareness that they are capable and meaningful to their families and others and that they must be. The results of the present study suggest that life purpose and ikigai are closely related—it supports the hypothesis that life purpose and ikigai are related, as suggested in previous studies.

In a previous study of a similar concept of ikigai, Kumano [22] listed subjective well-being as one of the reasons for living. Subjective well-being is defined by the dimensions of positive emotions, negative emotions, and life satisfaction. Regarding the relationship between subjective well-being and life purpose, Taguchi et al. [23] reported that those who were clear about their meaning and purpose in life had higher subjective well-being than those who were unclear. They also reported that all of the people with clear meaning and purpose in life had specific goals, while many of them with unclear goals did not. In other words, high life purpose leads to a positive life, which may affect subjective well-being. Thus, although some previous studies upheld the strength of the relationship between life purpose and subjective well-being, which is a similar concept to ikigai, no studies, including our present one, looked into the relationship between high life purpose and ikigai using an evaluation index of ikigai developed independently in Japan.

Therefore, in addition to the current preventive measures, it is necessary to take into account life purpose to support older people to achieve ikigai and self-actualization, along with ways to improve QOL. Moreover, the Manual for the Prevention of Long-Term Care states that individual services, such as the improvement of locomotor functions, are merely a means to achieve a goal and should not become a self-purpose [1]. Therefore, it is important to understand and support the life purpose of each subject.

As a concrete approach, we can give subjects time to think about the goals of their life and set long-term and short-term goals to realize their life goals, then have them participate in preventive exercises. Additionally, by understanding the life purpose of participants, the organizer of the care prevention program can provide individualized support for life goal realization all around.

As to the limitations and challenges in this study, although age, gender, and exercise habits were used to adjust confounders, it is also necessary to adjust other confounders related to older people, such as information on socioeconomic status, including educational background and annual income, the number of years participating in long-term care prevention projects, and any history of underlying diseases. By doing so, we might explain the strength of the relationship between life purpose and ikigai. Next, in the present study, the presence or absence of an exercise habit did not affect the high or low life purpose or ikigai. In this regard, even if a person has an exercise habit in daily life if the content of the exercise is not appropriate, it may not have a positive effect on the physical and mental aspects. Ito et al. [24] found that exercise interventions, not basic exercises such as strength training and stretching, but applied exercises such as stair climbing, daily living activities, and QOL-conscious exercises, have more positive effects on physical and mental health. In the present study, it was necessary to ask not only whether or not the subjects had an exercise habit, but also what kind of exercise they did and at what intensity and frequency. Finally, by conducting follow-up surveys according to the level of life purpose, it is possible to examine how the level of life purpose affects ikigai and changes in the state of care required.

## 5. Conclusions

Thus, in the multivariate analysis of high and low life purpose, only ikigai was found to be related. In the multivariate analysis of ikigai, life purpose was also the most relevant, indicating that life purpose and ikigai are strongly interrelated. The results suggest that to support older people to achieve ikigai and to improve QOL, it is necessary to take into an account life purpose in addition to the current nursing care prevention efforts.

## Figures and Tables

**Table 1 geriatrics-06-00073-t001:** Comparison between groups for each value in the high and low life purpose groups.

Evaluation Items	High Group (*n* = 39)	Low Group (*n* = 42)
Age (years)	77.9 ± 5.5	76.4 ± 4.9
Gender (persons)	males: 11, females: 28	males: 12, females: 30
Exercise habits (persons)	Yes: 27, No: 12	Yes: 26, No: 16
Ikigai (points)	33.9 ± 6.9 ***	27.3 ± 6.0
Physical functioning (points)	41.4	40.6
Physical role (points)	44.2	38.0
Bodily pain (points)	44.7	37.6
General health (points)	42.9	39.2
Vitality (points)	45.3	37.0
Social functioning (points)	44.0	38.2
Emotional role (points)	45.3	37.0
Mental health (points)	47.0 *	35.4

Mean ± Standard deviation; *t* test, χ^2^ test, Mann-Whitney *U* test, * *p* < 0.05, *** *p* < 0.001.

**Table 2 geriatrics-06-00073-t002:** Results of logistic regression analysis on high and low life purpose.

Variables	Odds Ratio	95% Confidence Interval	
Ikigai	1.156	1.058–1.264	**
Mental Health	1.049	0.951–1.156	

Binomial logistic regression analysis (forced entry method), **: *p* < 0.01.

**Table 3 geriatrics-06-00073-t003:** Results of logistic regression analysis on high and low life purpose with age, gender, and exercise habits added as variables.

Variables	Odds Ratio	95% Confidence Interval	
Ikigai	1.179	1.078–1.289	***
Age	1.015	0.913–1.129	
Gender	1.839	0.532–6.358	
Exercise habits	0.660	0.223–1.958	

Binomial logistic regression analysis with “age”, “gender”, and “exercise habit”; as independent variables (forced entry method), ***: *p* < 0.001.

**Table 4 geriatrics-06-00073-t004:** Results of Multiple Regression Analysis on Ikigai.

Variables	Standardized Partial Regression Coefficient	Standard Error	
Life purpose	0.434	0.433	***
Age	0.187	0.126	*
General health	0.046	0.114	
Vitality	0.076	0.151	
Social functioning	0.153	0.152	
Emotional role	0.134	0.168	
Mental health	0.221	0.158	
R^2^		0.430	

Multiple regression analysis (forced entry method) with “Ikigai” as the dependent variable and “life purpose”, “age”, “general health”, “vitality”, “social functioning”, “role emotional”, and “mental health” as independent variables, * *p* < 0.05, *** *p*< 0.001.

## Data Availability

No new data were created or analyzed in this study. Data sharing is not applicable to this article.

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
