# Peer review of "The Effect of High and Low Life Purpose on Ikigai (a Meaning for Life) among Community-Dwelling Older People—A Cross-Sectional Study"

_geriatrics, 2021, doi:10.3390/geriatrics6030073_

Round 1

Reviewer 1 Report

A re-evaluation of the translation is absolutely necessary, with the grammar check.

For example insted of "Yhe purpose of this study is [...]" (line 86),correct is "The purpose of this study is [...]".

Also, at Line 197, chapter 4. Discussion “A The results in the univariate analysis for mental health and life purpose showed “, I don’t think “ A The results” it’s a correct English version.

I think it would be useful to drop table 2 because it does not bring enough data for a table. The information can be included in the text, considering that it has only one row.

The same remark for table 2, whose information can be included in the text or can be further developed in its current form by adding several rows with other information.

The conclusions may be more developed.

More attention to bibliography, it can be improved with more important data from the literature, on the published topic.

Reviewer 2 Report

Dear authors,

This is an interesting analysis of high and low life purpose and ikigai. The authors found that life purpose is related to ikigai and suggest that this be taken into consideration in the context of the care of older people. Despite the small sample size, the methods and results are appropriate. The manuscript requires significant editing for typographical errors and to improve the structure and flow before being suitable for publication. The abstract and introduction, in particular, require careful editing. The strength of the paper lies in the discussion. Some examples and comments are as follows:

Title: This needs to be changed as I am not sure that “elderlies” is a word. Nevertheless, please avoid use of the word elderly and elderlies as it is no longer appropriate, and instead use older people or older adults.

Abstract does not state what Ikigai is.

Abstract Line 12, Please do not start sentence with a number.

Please include some of the results in the abstract.

Introduction:

Please inform the reader what country the MHLW is located in.

There are spelling errors and typos throughout.

Methods:

In the introduction and methods, the paragraph structure is inconsistent. For example on lines 99-105, it is not clear why this is not one paragraph.

Please use sub-headings in the methods such as following STROBE or similar.

Discussion:

Line 242: Person instead of subject.

Reviewer 3 Report

One of the weaknesses of the research is related to the scarcity of cited bibliographic sources. I think that the topic covered in the article is still in an early stage.

Round 2

Reviewer 1 Report

The current version is much improved. I consider that you
answered at all my observations. This English translation
is clearly superior to the initial version of the article. I believe that the article can be published in its current form.

Reviewer 2 Report

The authors have addressed my comments.